# Impact of Perch Provision Timing on Activity and Musculoskeletal Health of Laying Hens

**DOI:** 10.3390/ani14020265

**Published:** 2024-01-15

**Authors:** Mallory G. Anderson, Alexa M. Johnson, Cerano Harrison, Mireille Arguelles-Ramos, Ahmed Ali

**Affiliations:** 1Department of Animal and Veterinary Sciences, Clemson University, Clemson, SC 29634, USA; mga5@clemson.edu (M.G.A.); ajohn43@clemson.edu (A.M.J.); ceranoh@clemson.edu (C.H.); marguel@clemson.edu (M.A.-R.); 2South Carolina Translational Research Improving Musculoskeletal Health Center, Clemson University, Clemson, SC 29634, USA; 3Animal Behavior and Management, Veterinary Medicine, Cairo University, Cairo 12613, Egypt

**Keywords:** laying hen, perch provision, musculoskeletal health, activity level

## Abstract

**Simple Summary:**

In this study, we investigated the enduring impacts of perch provision timing on the musculoskeletal health of laying hens. A total of 810 pullets experienced different housing conditions: continuous access to multi-tier perches from 0 to 40 weeks (CP), no perch access (NP), early access during the rearing phase from 0 to 17 weeks (EP), or solely during the laying phase from 17 to 40 weeks (LP). Monitoring from week 24 to 40 included individual activity levels, blood sample collection for bone demineralization markers, and euthanasia for computed tomography scans at 40 weeks. Results showed that hens with continuous perch access demonstrated higher overall activity at 24 weeks and improved musculoskeletal health at 40 weeks compared to those with no access. Late perch access also positively affected activity, muscle deposition, and bone strength. Conversely, early access did not yield long-term impacts on activity or musculoskeletal health except for intermediate responses in bone demineralization. These findings highlight the importance of timing in perch provision, emphasizing that continuous or late access enhances the well-being and musculoskeletal health of laying hens in comparison to no access at all. Early access to perches did not have a long-term beneficial effect on the activity or musculoskeletal health of laying hens. The study suggests that optimizing perch exposure timing can contribute to sustained improvements in the physical condition of laying hens throughout their reproductive lifespan.

**Abstract:**

Laying hens can experience a progressive increase in bone fragility due to the ongoing mobilization of calcium from bones for eggshell formation. Over time, this escalates their susceptibility to bone fracture, which can reduce their mobility and cause pain. The provision of perches as an exercise opportunity could potentially enhance bone strength, but the timing of exposure to perches during the birds’ development may modulate its impact. The objective of this study was to investigate the enduring impacts of perch provision timing on the musculoskeletal health of laying hens. A total of 812 pullets were kept in different housing conditions (seven pens/treatment, 29 birds/pen) with either continuous access to multi-tier perches from 0 to 40 weeks of age (CP), no access to perches (NP), early access to perches during the rearing phase from 0 to 17 weeks of age (EP), or solely during the laying phase from 17 to 40 weeks of age (LP). At weeks 24, 36, and 40 of age (n = 84 birds/week), three birds per pen were monitored for individual activity level, and blood samples were collected from a separate set of three birds per pen to analyze serum levels of tartrate-resistant acid phosphatase 5b (TRACP-5b) and C-terminal telopeptide of type I collagen (CTX-I) as markers of bone demineralization. At 40 weeks of age, three birds per pen (n = 84) were euthanized for computed tomography scans to obtain tibial bone mineral density (BMD) and cross-sectional area (CSA) with further analysis including muscle deposition, tibial breaking strength, and tibial ash percent. During week 24, hens from CP, EP, and LP pens had the highest overall activity compared to hens from NP pens (*p* < 0.05) with no differences between treatments for overall activity level during weeks 36 or 40 (*p* > 0.05). During weeks 24, 36, and 40, hens from CP and LP pens showed greater vertical and less horizontal activity compared to hens from EP and NP pens (*p* < 0.05). TRACP-5b and CTX-I concentrations did not differ between treatments at week 24 of age (*p* > 0.05). Hens from CP pens had the lowest TRACP-5b and CTX-I concentrations at 36 weeks of age with EP and LP hens showing intermediate responses and NP hens having the highest concentration (*p* < 0.05). At 40 weeks of age, CP hens had the lowest TRACP-5b and CTX-I concentrations compared to NP hens (*p* < 0.05). Total bone CSA did not differ between treatments (*p* > 0.05), but CP had greater total BMD than NP (*p* < 0.05) with no differences between EP and LP treatments. CP and LP hens had larger biceps brachii, pectoralis major, and leg muscle groups as well as greater tibial breaking strengths than EP and NP treatments (*p* < 0.05). CP hens had higher tibial ash percentages compared to EP, LP, and NP (*p* < 0.05). Our results indicate that providing continuous perch access improves the musculoskeletal health and activity of laying hens at 40 weeks of age compared to no access and that late access to perches has a beneficial impact on activity, muscle deposition, and bone strength.

## 1. Introduction

A major welfare concern in the laying hen industry is osteoporosis, which refers to the progressive decrease in structural bone, leading to increased susceptibility to bone fractures [1,2]. As pullets reach sexual maturity, osteoblasts begin forming medullary bone, which is intended as a reliable source of calcium for eggshell formation [1]. During calcium mobilization, osteoclasts resorb both medullary and structural bone so that over time, the hen remains reproductively active; there is a progressive decrease in structural bone, resulting in bone fragility [1,2]. The increase in bone fragility increases the susceptibility of the bones to fracture [2]. Fractures can reduce hen mobility and cause acute or chronic pain [3,4]. Solutions to reduce the occurrence of osteoporosis by improving bone strength in laying hens include dietary interventions [5,6,7,8,9], genetic selection for bone quality [10,11,12,13] and providing perches as an opportunity for exercise to increase activity levels [14,15,16,17,18,19].

The type of housing system can impact the activity level and musculoskeletal health of laying hens. For example, hens housed in conventional cages lack the opportunity for exercise and may be more susceptible to osteoporosis compared to hens housed in alternative systems [2,12,16]. In the United States, while the inclusion of perches within poultry housing is not compulsory, it is noteworthy that their incorporation is recommended according to guidelines provided by the United Egg Producers, contrasting with the regulatory framework in the European Union, where the provision of perches is obligatory. Providing more opportunities for load-bearing exercise, such as incorporating multi-tier perches into alternative housing systems, can increase hen activity and improve musculoskeletal health [12,15]. Laying hens are highly motivated to perch, mainly as a means of defense from predators or to avoid aggression from conspecifics [20]. Hens showed frustration-related behaviors when deprived of access to perches [19,20]. Moreover, the ability to perch is important in laying hens reared in aviary systems, where resources (such as food, water, and nest boxes) can be provided in different vertical levels of the system. Perching is a highly motivated behavior that can be used to stimulate activity, improve musculoskeletal health, and reduce the occurrence of osteoporosis in laying hens [19,21,22].

With an increased freedom for movement and exercise comes an increased risk for keel bone injuries, which up to 80% of laying hens may experience [23,24]. Keel bone injuries are a complex welfare problem with a multitude of interplaying risk factors such as genetics, nutrition, and environment [25,26]. Keel bone fractures can frequently occur due to falls or collisions with furniture within complex housing environments or also due to other short-duration traumatic events [27,28]. In contrast, keel bone deformities typically occur from prolonged mechanical pressure load during perching [29]. The exact factors influencing keel bone injuries in laying hens are complex, and the extent to which physical activity at certain ages plays a role in susceptibility to keel bone damage is unknown [24,25].

A major contributing factor influencing a hen’s ability to perch is the age during which they are exposed to perches. Without prior experience with perches, young chicks show a poor ability to use perches later in life [17,27]. Early access to perches may provide pullets with proper exposure to and practice perching, increasing muscle mass and bone strength. Perch access during rearing may improve the birds’ ability to use perches better later in life and also result in stronger musculoskeletal systems at the start of lay, reducing the risk of osteoporosis as an adult. For example, some previous studies found the benefits of rearing pullets in alternative housing systems on bone composition and strength at 16 weeks of age [18,28], where the beneficial impact on pullet bone composition observed at 16 weeks of age continued during the laying phase [29]. Furthermore, rearing pullets in conventional cages with perches resulted in some benefits to bone health in 71-week-old hens [15]. Rearing in alternative housing systems has also been shown to reduce keel bone damage during the laying phase [30,31]. By focusing on strengthening bones during development, we may observe hen bones that are better equipped to handle the mobilization of calcium in a proactive approach to prevent osteoporosis.

This study aimed to observe the long-term effects of perch provision timing, either during only the rearing phase, only the laying phase, both phases, or neither on the musculoskeletal health and activity of adult laying hens. Pullets were reared either with or without access to multi-tier perches until 17 weeks of age, at which point half of the pullets with perches transitioned to pens without perches, and half of the pullets without perches transitioned to pens with perches. We hypothesized that pullets with continuous access to perches would show improved musculoskeletal health and increased activity compared to pullets without access to perches, and pullets with perch access during only the rearing or only the laying phase showing intermediate responses.

## 2. Materials and Methods

### 2.1. Ethics

This experiment was approved by Clemson University’s Institutional Animal Care and Use Committee (protocol #AUP2021-0068).

### 2.2. Animals and Housing

This experiment was conducted at Clemson University’s poultry facility in South Carolina, USA. Day-old Hy-Line brown chicks (n = 840) were randomly allocated across 30 pens (28 birds/pen) until 17 weeks of age. Pens (5.2 m^2^) contained 7.6 cm of clean pine wood shavings as bedding. Trough feeders were provided for the first 3 weeks of age at which point hanging feeders were used. Birds had ad libitum access to water and feed. From 0 to 3 weeks of age, feed was provided in tube feeders and water in gallon drinkers. For the first week of life, supplementary feed trays were provided. After 3 weeks, feed was provided in circular hanging feeders and water was available in automatic cup drinkers. For the first 3 weeks of age, heat was provided by one focal electric brooder per pen and a gas-fired brooder for the entire house. The temperature was initially set at 35–36 °C at day 0; then, it progressively reduced by 2–4 °C every week until 3 weeks of age when brooders were removed. Temperature was reduced weekly until 6 weeks of age to 21 °C, and then they were maintained until the end of the study, following the standard breed guidelines [32]. Light was provided by one 60-watt incandescent overhead lightbulb per pen, and each pen was kept on a decreasing light schedule starting at 20 L:4D during the first week and was decreased by increments of either 1.5 or 3 h until 10 L:14D from 7 weeks of age until the end of the study when birds were 40 weeks old [32].

### 2.3. Treatments

From 0 to 17 weeks of age, 15 pens (420 birds) were provided with perches while the remaining 15 pens (420 birds) were without perches. At 17 weeks of age, half of the birds with perches transitioned to pens without perches, and half of the birds without perches transitioned to pens with perches until 40 weeks of age. This resulted in four treatment groups (7 pens/treatment and 28 birds/pen after accounting for mortality during weeks 0–17): continuous perch (CP; perch access from 0–40 weeks of age), no perch (NP; no perch access from 0–40 weeks of age), early perch (EP; perch access from 0–17 weeks of age), and late perch (LP; perch access from 17–40 weeks of age). The perch structure was constructed to be adjustable with perch rungs made of 5 × 5 cm pressure-treated wooden lumber. Each perch structure contained 3 rungs of varying height, each 165.1 cm in length, resulting in 495.3 cm of total perch space and approximately 19 cm of perch space per bird. In the CP group, rung heights and distance between rungs were gradually increased concurrently with the growth of the birds to ensure they were easily accessible. For the first 11 days of age, the 3 rungs were 15.2 cm, 22.8 cm, and 30.4 cm high off the ground (Figure 1a). For the next 8 days, the 3 rungs were 22.8 cm, 38.1 cm, and 54.6 cm high off the ground (Figure 1b). The perch rungs were altered once more on day 20 of age to 38.1 cm, 62.2 cm, and 88.4 cm high with a 12.7 cm distance between each perch rung (Figure 1c).

### 2.4. Activity

Bird activity was monitored using an accelerometer over 3 consecutive days during weeks 24, 36, and 40 of age (n = 84 birds/week). At each time point, 3 birds/pen were caught after the lights went off. Birds were selected from among different resources and perch levels in an attempt to sample hens that were representative of the flock in that pen. Birds were fitted with a harness that was used to secure the accelerometer. An acceleration data logger (Onset HOBO PendantG acceleration data loggers, Onset Computer Corporation, Bourne, MA) was inserted inside each harness. The loggers used in the current study were 58 × 33 × 23 mm in size and 16 g in weight with a ±3 g; 29.4 m/s^2^ measuring range and ±0.105 g; 1.03 m/s^2^ accuracy level when operating between −20 and 70 °C. Loggers were oriented on the hens, so the *X*-axis captured forward and backward movement (craniocaudal movement), the *Y*-axis captured sideways movement (mediolateral movement), and the *Z*-axis captured vertical movement (dorsoventral movement) of the hens. Loggers were firmly secured inside the harness to reduce noise in the data due to the movement of the loggers themselves and to prevent changes in logger orientation. After fitting focal birds with harnesses and accelerometers, hens were given 1 day to habituate to wearing the equipment. During this period, hens were monitored to ensure that vests were not impacting behavior and locomotion abilities. After acclimation, loggers recorded hens’ movement across 3 consecutive days (72 h) at each time point with a scanning frequency of 20 Hz (±3 g to +3 g) in 3 axes.

### 2.5. Musculoskeletal Health

#### 2.5.1. Computed Tomography (CT) Image Acquisition

At 40 weeks of age, 3 birds per pen (n = 84) were euthanized on-farm by CO_2_ inhalation, placed in a cooler of ice, and immediately transported to the Godley-Snell Research Center on Clemson University’s campus. Upon arrival, birds were individually placed inside a V-shaped foam cradle in a dorsal recumbent position atop a hydroxyapatite calibration phantom (QRM Quality Assurance in Radiology and Medicine, Möhrendorf Germany). The head and the legs of the bird were extended in opposite directions and were taped to maintain this positioning in the foam cradle during image acquisition. CT images were acquired using a helical mode, head 0–10 kg protocol, 0.5 mm slice thickness, and bone and soft tissue reconstruction algorithms. CT images were acquired using a Toshiba Aquilion TSX-101A, 16-slice scanner (https://www.gehealthcare.com, accessed on 16 November 2023, GE Healthcare, Chicago IL, USA), a single bird scan and image construction required approximately 7 min. Birds were dissected immediately after CT scanning and frozen at −20 °F for further testing.

#### 2.5.2. Tibiotarsal CT Image Analyses

For each CT study, measurements of the right tibiotarsal bone and muscle were made using a standardized CT image analysis protocol previously published by [33]. Cross-sectional density (HU) and area (mm) of the total and medullary components of the tibiotarsal bone were recorded at predefined proximal, middle, and distal transverse slice locations using hand-traced regions of interest (Figure 2a,b). The cross-sectional area (CSA) of the muscle group surrounding the tibiotarsus at each of the predefined proximal, middle, and distal locations was also measured. The CT densities for each of the rods in the bone calibration phantom were recorded using the oval ROI tool (Figure 2c). The CT densities in HU were then converted to hydroxyapatite values using graphical analysis techniques described in [33].

#### 2.5.3. Muscle Deposition

After CT scanning, birds (n = 84) were prepared for dissection, and the separation of muscles was conducted following the procedures described by [18] and with the assistance of a veterinarian (A.A.) to ensure consistent muscle specimen collection. Birds were opened by cutting the skin on the caudal tip of the keel bone and peeling it back to expose the interior of the bird. To remove the right bicep and triceps brachii, the skin of the wing was peeled back, and a blunt dissection was made along the line of demarcation between the biceps and triceps. The bicep and triceps were gently freed from the bone, and the proximal and distal tendons were cut at the bone level. To separate the pectoralis muscles, fascia was cut along the line of demarcation, separating the fats from the pectoralis muscles and severing all the attachment at the origin (crania sternum, furcula, and sternal ribs), and at the insertion of the major (proximal ventral surface of the humerus) and of the minor (proximal dorsal surface of the humerus). The left leg muscles, tendons, and ligaments were detached from the bone, the Achilles tendon was severed, and the fascia along the synsacrum was detached. All muscles were immediately weighed upon removal. The left tibiae were frozen at −20 °C for ash percentage, and the right tibiae were frozen at −20 °C for breaking strength measures.

#### 2.5.4. Tibia Breaking Strength

Mechanical properties of the right tibiotarsi were assessed using a three-point bending test as specified by the American National Standards Institute (ANSI) standards for the application of 3-point bending on animal bones [34]. Testing was performed using an Instron Dynamic and Static Material Test system (Model 5944, Instron Corp., Canton, MA, USA) equipped with a 500 N load cell and Automated Material Test System software (8800 MT Controller, Instron Corp., Canton, MA, USA). Prior to testing, previously frozen legs were thawed at refrigerator temperature. Muscles surrounding the tibiotarsus were carefully dissected, tibiotarsal length and diameter at the midpoint were recorded, and the bones were wrapped in saline-soaked paper towels until testing to prevent the bones from drying out.

Rounded support pins and breaking blade were manufactured based on ANSI standards for the application of 3-point bending on animal bones [34]. A furculum width of 4 cm was used. This width did not adhere to the ANSI standards but was decided upon based on a consensus among co-authors. Due to the anatomy of the laying hen tibiotarsus, a 4 cm width ensured that the tibiotarsus was able to rest on the furculum in a manner in which the load would be applied to the midpoint of the bone evenly in the craniocaudal plane. The crosshead speed used was 3 mm/min, and the test was carried out to failure. Load and displacement data were collected and were used to obtain the breaking strength (N), stiffness (N/mm), and maximum bending moment (N/m).

#### 2.5.5. Tibia Ash Percentage

The left tibiotarsi of euthanized birds was thawed approximately 24 h prior to data collection. The bones were cleaned from any surrounding muscles and soft tissues, and tibiae were separated from the fibula. The tibiae were cut into 3 pieces to fit into a Soxhlet chamber for ether extraction. Ceramic crucibles were air-dried for one hour and then placed in a desiccator for another hour. The weight of the dried crucibles was recorded. Left tibiae were dried at 100 °C for one hour, placed in a desiccator for another hour, and their weight was recorded. Tibiae were then placed inside the dried ceramic crucibles and ashed (ashing oven: Thermolyne 30400, Barnstead International, Dubuque, IA, USA) for 6 h at 600 °C. The ash was placed in a desiccator for one hour, and then the ash weight was recorded. The percentage of tibia ash was calculated by dividing the tibia ash weight by the tibia dry weight and multiplying by 100.

#### 2.5.6. Bone Resorption Markers

During weeks 24, 36, and 40 of age, blood samples were collected from the brachial wing vein of 3 birds/pen (n = 84/week). Whole blood samples were transferred to 1.5 mL Eppendorf tubes, and serum was separated at 6000 rpm for 10 min at 4 °C. In order to test for the occurrence of bone resorption, serum samples were analyzed for levels of tartrate-resistant acid phosphatase 5b (TRACP-5b) and C-terminal telopeptide of type I collagen (CTX-I) using commercial ELISA kits Nanjing Jiancheng Institute of Bioengineering (Nanjing, China) and MyBioSource (San Diego, CA, USA), and according to manufacturer’s instructions.

#### 2.5.7. Data Processing and Statistical Analysis

The raw accelerometer data, consisting of the date, time, and the related impulse in the X, Y, and Z dimensions, were downloaded from the devices (HOBOware Graphing & Analysis Software 001, Onset, Bourne, MA, USA) at the end of each 3D observation period. Data on hens’ vertical (*a_z_*: dorsoventral movement across vertical levels), horizontal (*a_x_*: craniocaudal movement within the same vertical level), and lateral movement (*a_y_*: mediolateral movement within the same vertical level) during light hours were obtained directly from loggers. Hens’ triaxial movement (*A_s_*) was calculated by summing and averaging raw movement data as follows.
$$A_{s}=\sqrt{a_{x}^{2}+a_{y}^{2}+a_{z}^{2}}$$

Acceleration data (gravity “g”) were post-processed using MATLAB (MATLAB and Statistics Toolbox Release 2012, The MathWorks, Inc., Natick, MA, USA). In order to accurately calculate the incidence of massive acceleration shifts on the vertical (z) axis that represents perching, data were smoothed from noisy components by removing all minor acceleration fluctuations using a loop function.
$$A_{i}=\frac{1}{3}\sum_{j=i-1}^{i+1}A_{j}A_{i}^{\prime }=\left\{\begin{array}{c} \mu ,\;if\;\left|A_{i}-\mu \right|<t\\ \mu ,\;if\;\left|A_{i}-\mu \right|\geq t \end{array}\right.$$

Data smoothing included the passing of the raw acceleration values (*A_j_*) through an asymmetrical 3-point moving average low-pass filter (*i* = the middle point in the 3-point-moving average low-pass filter) and through a step function to define thresholds used to remove minor fluctuations (*t* = threshold values of minor fluctuations, i.e., between 0.001 and 0.043 g). After processing data, perching events were recognized by detecting massive shifts in acceleration in the *z*-axis of activity. which was defined as incidence (frequency “F”) of perching or “vertical displacement”. In order to precisely detect acceleration shifts due to perching and define thresholds for minor fluctuations in the *z*-axis, timestamped videos of birds while perching were obtained and compared with the corresponding activity data. Using the approach enabled us to locate shifts in *z*-axis acceleration mainly caused by perching and define the threshold cutoff points to remove minor fluctuation.

Data were analyzed using the R software (version 3.3.1) with the package “stats” (R Core Team, 2013). To test for the main effects of treatment (CP, EP, LP, and NP) and the age of the birds (activity and bone demineralization: 24, 36, and 40 weeks) on each variable, generalized linear mixed-effects models (GLMMs) were conducted using the “lme4” package [35]. In each GLMM, the interaction term between main effects was also tested as a fixed effect, and bird ID, pen, and day for activity were tested as random effects, with the family set to “Quasibinomial” for proportion data (ash %) and “Poisson” for the other data. Tukey’s HSD multiple comparison procedure was used for post hoc comparisons using the “multcomp” package [36]. The “DHARMa” package was used for proportion data (ash%) to test residual distribution and assumptions for GLMM, while the Shapiro–Wilk test was utilized (i.e., activity (g), breaking strength (N), stiffness (N/mm)) for the normality analysis of the model residuals. Statistical significance was set at *p* < 0.05. Descriptive statistics were calculated using the “psych package”, and data are presented as mean ± standard error of the mean (SEM).

## 3. Results

### 3.1. Activity

During week 24, hens from CP, EP, and LP exhibited the greatest amount of overall activity compared to hens from NP pens (*p* = 0.021, 0.033, 0.036, respectively; Table 1). There were no differences between treatments for overall activity levels during weeks 36 or 40 (Table 1). During all observation weeks, hens from CP and LP pens showed greater vertical activity (week 24: (CP: *p* = 0.019, 0.023; LP: *p* = 0.026, 0.031); week 36: (CP: *p* = 0.025, 0.0.35; LP: *p* = 0.038, 0.029); week 40: (CP: *p* = 0.028, 0.031; LP: *p* = 0.032, 0.028); Table 1), and less horizontal activity (week 24: (CP: *p* = 0.033, 0.029; LP: *p* = 0.034, 0.027); week 36: (CP: *p* = 0.028, 0.0.33; LP: *p* = 0.037, 0.039); week 40: (CP: *p* = 0.032, 0.027; LP: *p* = 0.033, 0.037); Table 1) compared to hens from EP and NP pens. During all observation weeks, hens from CP and LP pens exhibited a higher average daily vertical displacement per bird compared to hens from EP and NP pens (week 24: (CP: *p* = 0.018, 0.021; LP: *p* = 0.023, 0.019); week 36: (CP: *p* = 0.022, 0.027; LP: *p* = 0.023, 0.031); week 40: (CP: *p* = 0.022, 0.023; LP: *p* = 0.019, 0.027); Table 1). There were no differences across weeks within the same treatment (*p* > 0.05).

### 3.2. Tibial Bone Mineral Density (BMD) and Cross-Sectional Area (CSA)

There were no differences between treatments for total CSA (Table 2). CP hens had greater cortical CSA and cortical BMD at all locations than other treatment groups (cortical CSA: proximal (*p* = 0.022, 0.018, 0.029), middle (*p* = 0.022, 0.031, 0.024), distal (*p* = 0.031, 0.028, 0.021); cortical BMD: proximal (*p* = 0.022, 0.027, 0.036), middle (*p* = 0.024, 0.023, 0.025), distal (*p* = 0.031, 0.019, 0.031); Table 2). EP and LP hens had greater cortical CSA at the proximal (*p* = 0.021, 0.023, respectively), middle locations (*p* = 0.032, 0.028, respectively), and greater cortical BMD values at all locations than NP hens (proximal (*p* = 0.019, 0.024), middle (*p* = 0.023, 0.025), distal (*p* = 0.027, 0.035); Table 2). However, CP hens had greater total BMD at all locations than NP hens (proximal: *p* = 0.013, middle: *p* = 0.009, distal: *p* = 0.012; Table 2), and at the middle (*p* = 0.029, 0.036, respectively) and distal (*p* = 0.035, 0.036, respectively) locations than EP and LP, while EP and LP had greater total BMD at all locations than NP (proximal: *p* = 0.019, 0.023; middle: *p* = 0.022, 0.031; distal: *p* = 0.029, 0.036, respectively; Table 2).

### 3.3. Muscle Deposition

Hens from CP and LP pens had heavier biceps brachii (CP: *p* = 0.032, 0.025; LP: *p* = 0.036, 0.029; Table 3), pectoralis majors (CP: *p* = 0.0.026, 0.026; LP: *p* = 0.027, 0.037; Table 3), and leg muscle groups (CP: *p* = 0.031, 0.036; LP: *p* = 0.029, 0.027; Table 3) compared to hens from EP and NP pens. There were no differences between treatments for weights of the triceps brachii or pectoralis minor (*p* > 0.05; Table 3).

### 3.4. Tibia Breaking Strength

At week 40 of age, housing hens in CP and LP pens resulted in greater tibia-breaking strengths (CP: *p* = 0.019, 0.011; LP: *p* = 0.017, 0.009, respectively) and stiffness (CP: *p* = 0.021, 0.013; LP: *p* = 0.006, 0.019, respectively) compared to housing hens in EP and NP pens (Figure 3).

### 3.5. Tibia Ash Percentage

At week 40 of age, the tibia of hens housed in CP pens contained a higher ash percentage compared to hens housed in EP, LP, and NP pens (*p* = 0.003, 0.009, 0.012, respectively; Figure 4).

### 3.6. Bone Demineralization

There were no differences in TRACP-5b or CTX-I concentrations between treatment groups at week 24 of age (*p* > 0.05; Figure 5a,b). At 36 weeks of age, hens from CP pens had the lowest TRACP-5b concentration compared to other groups (EP: *p* = 0.039, LP: *p* = 0.023, NP: *p* = 0.013), which was followed by hens from EP pens (LP: *p* = 0.036, NP: *p* = 0.023), then LP pens (NP: *p* = 0.046), with hens from NP pens having the highest concentration (Figure 5a). Furthermore, hens from CP and EP pens had the lowest CTX-I concentrations at week 36 of age compared to hens from LP and NP pens (CP: *p* = 0.011, 0.019; EP: *p* = 0.036, 0.037, respectively; Figure 5b). At week 40 of age, CP hens had the lowest TRACP-5b (EP: *p* = 0.036, LP: *p* = 0.035, NP: *p* = 0.026) and CTX-I (EP: *p* = 0.029, LP: *p* = 0.031, NP: *p* = 0.036) concentrations compared to EP and LP hens, with NP hens having the highest concentrations (CP: *p* = 0.026, EP: *p* = 0.023, LP: *p* = 0.013; Figure 5a,b).

## 4. Discussion

### 4.1. Activity

Perch access did impact overall hen activity level at week 24 of age but not at weeks 36 and 40, with hens from CP, EP, and LP pens showing the greatest amount of overall activity compared to hens from NP pens. Interestingly, access to perches from 0 to 17 weeks of age resulted in increased overall activity at 24 weeks of age. However, we did not see an effect of perch access on overall activity at 36 and 40 weeks of age. During weeks 24, 36, and 40 of age, hens from CP and LP pens performed more vertical activity, less horizontal activity, and had a higher average daily vertical displacement per bird compared to hens from EP and NP pens. This is likely due to the fact that hens from CP and LP pens had access to an appropriate perching structure and thus more opportunities to move vertically compared to EP and NP hens who had no access to perches and could not physically move vertically to the same extent. Because hens are highly motivated to perch on high areas of their home pen, it follows that the hens with access to multi-tier perches showed more vertical activity, as they were likely jumping to reach elevated surfaces within the pen [14,15,17,37]. Furthermore, EP hens exhibited a greater proclivity to perch on elevated structures, such as feeders and nest boxes, within their surroundings in comparison to NP hens. This anecdotal observation suggests an increased inclination to perch among EP hens, which was potentially attributed to their early exposure to perching experiences during the rearing phase.

### 4.2. Tibial Bone Mineral Density (BMD) and Cross-Sectional Area (CSA)

Perch access influenced the tibial bone mineral density at 40 weeks of age, with birds from CP pens having a greater total BMD content compared to hens from NP pens, indicating access to perches beneficially impacted tibial bone mineral density. Furthermore, CP hens had greater cortical CSA and BMD at all locations than the other treatment groups with EP and LP hens having greater CSA and BMD values than NP hens. This finding is in line with previous studies that found access to perches increases bone strength in laying hens [16,38,39]. Ours results align with previous research and suggest that load-bearing exercise from continual perch use improves bone characteristics with early or late perch access having intermediate responses compared to no perch access at all. However, one previous study found that the addition of perches to conventional cages did not increase the tibial BMD of 71-week-old hens [15]. This could be due to differences in the design of perches or that White Leghorns were the strain of the birds used in the previous study compared to Hy-Line Brown in the current study. The minimal differences noted in tibial BMD and (CSA) between EP and LP hens are surprising. This suggests that even in the later stages of the laying cycle, hens can derive benefits from perch provisions, thereby enhancing their bone health. On the other hand, early provision of perches during the rearing phase appears to have a positive impact on bone health throughout the laying phase. However, it is important to note that neither scenario is directly compared with the potential benefits of providing perches during both rearing and laying phases.

### 4.3. Muscle Deposition

By the end of the study at 40 weeks of age, hens from CP and LP pens had heavier biceps brachii, pectoralis majors, and leg muscle groups compared to hens from EP and NP pens with no differences for the weights of triceps brachii or pectoralis minors. Providing continuous access to perches resulted in heavier muscles compared to not providing perches at all due to higher activity levels during rearing and laying. Perching is considered a form of load-bearing exercise that has previously shown to increase muscle deposition in poultry [14,15]. By the end of the study, hens with late access to perches had heavier muscles than hens with access to perches during the rearing phase. This was contrary to some previous work, as early access to perches has been shown to increase muscle deposition in adults due to there being more opportunities for exercise during development [15,40]. However, hens with access to perches during the lay phase performed more vertical activity and jumped more frequently than hens without access to perches, suggesting these activities beneficially impacted muscle growth even after puberty. Access to perches during the lay phase ultimately had a more beneficial effect on muscle deposition at 40 weeks of age compared to access to perches during the rearing phase.

### 4.4. Tibia Breaking Strength

Timing of perch access impacted tibia strength with CP and LP hens having a higher breaking strength and stiffness at 40 weeks of age compared to EP and NP hens. Some previous studies found no difference in tibia breaking strength between housing systems with or without perches [41,42]. However, other studies found that access to perches as an adult improves tibia strength: for example, hens housed with perches from 19 weeks of age had stronger bones and better preserved cortical bone than hens housed without perches at 65 weeks of age [16]. Furthermore, hens housed with perches from 16 weeks of age had a higher tibia breaking strength than hens in conventional cages at 73 weeks of age [21]. In agreement with our results, the previous study found no effect of rearing environment on adult bone breaking strength. However, in its companion study, they discovered a greater beneficial effect of rearing pullets with perches on breaking strength at 16 weeks of age than what was discovered for adult hens, highlighting the importance of providing opportunities for exercise during bone development [18]. Although we did not find an effect of providing perches during rearing on adult bone breaking strength, numerical differences between the CP (breaking strength: 311.02 N; stiffness: 289.88 N/mm) and LP (breaking strength: 289.96 N; stiffness: 256.26 N/mm) groups suggest that providing perches during the rearing (i.e., bone development) and lay phase may be more beneficial to bone strength than providing perches during the lay phase alone. Our results suggest that providing perches either continuously or at the beginning of the lay phase permits sufficient opportunity for exercise to improve breaking strength by week 40 of age compared to hens not provided perches at all or only during the rearing period.

### 4.5. Tibia Ash Percentage

Perch access impacted tibia ash percent at 40 weeks of age, where the tibia of hens housed in CP pens contained a higher ash percentage compared to hens housed in EP, LP, and NP pens, suggesting that continuous perch access (perch access from 0 to 40 weeks of age) beneficially impacted bone mineral content. One prior study found that free range hens with access to perches had a greater tibia ash percent at 38 and 45 weeks of age compared to hens in conventional cages with or without access to perches, indicating that a greater freedom of movement and more opportunities for exercise improve tibia mineral content compared to providing simple perches in a caged environment alone [43]. In agreement, hens housed in floor pens with perches had higher tibia ash percentages compared to hens housed in conventional cages [44]. However, a couple previous studies found no relationship between housing type and tibia ash percent [45,46]. In our study, continuous perch access improved tibia mineral content compared to early, late, or no access to perches.

### 4.6. Bone Resorption

In our study at 24 weeks of age, there were no differences in TRACP-5b and CTX-I levels between treatment groups, indicating all treatments started at similar levels of bone resorption. We observed differences in bone resorption at 36 weeks of age, with the lowest TRACP-5b concentrations found in hens from CP pens, which was followed by hens from EP pens, then LP pens, with hens from NP pens having the highest concentrations. Furthermore, hens from CP and EP pens had the lowest CTX-I concentrations at week 36 of age compared to hens from LP and NP pens. Our results indicate that hens from CP pens showed mild bone resorption compared to hens from NP pens, which showed the highest levels due to an absence of bone reservoirs. The lack of activity during rearing and laying does not improve bone characteristics and leaves adult laying hens at risk for increased bone resorption and ultimately weakened bones. Hens from EP pens showed low levels of bone resorption, which was comparable to hens from CP pens at week 24 of age. However, at week 36 of age, bone resorption increased to a level slightly higher than hens from CP pens, but it was still less than hens from LP and NP pens, which is an effect that can be contributed to a higher bone reserve due to increased perching activity during rearing. At week 40 of age, CP hens had the lowest TRACP-5b and CTX-I concentrations compared to EP and LP hens with NP hens having the highest concentrations. Both EP and LP hens showed similar bone resorption levels, as the effect of early perch access dissipated and the effect of later perch access slowed bone resorption levels compared to hens from NP pens.

## 5. Conclusions

The outcomes derived from our investigation indicate that the continuous provision of multi-tier perch access throughout the rearing and early lay phase (0–40 weeks of age) exerts a favorable influence on activity level and thus the musculoskeletal health of laying hens at 40 weeks, thereby contributing to an improvement in overall hen welfare when compared to the absence of perch access. Similarly, the availability of perches during the early lay period (17–40 weeks of age) demonstrates positive effects on activity, muscle deposition, and bone strength; however, these benefits are not as pronounced as those observed with continuous perch access. Moreover, the introduction of perches during the rearing phase (0–17 weeks of age) is associated with a deceleration in bone demineralization, aligning with the outcomes observed in hens with access to perches during the laying phase. Nevertheless, early perch access does not manifest an overarching positive impact on the musculoskeletal health or activity levels of laying hens at 40 weeks of age, suggesting that early exposure during developmental stages does not confer long-term benefits in these aspects. The findings underscore the need for further research to elucidate the effects of early exercise during the rearing phase on bone demineralization in adult laying hens, providing a more comprehensive understanding of the nuanced relationships between developmental experiences and musculoskeletal health.

## Figures and Tables

**Figure 1 animals-14-00265-f001:**
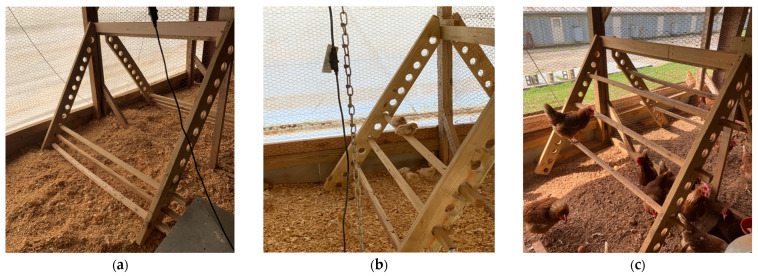
Perch and adjustable rung heights in continuous perch (CP) treatment groups during days (**a**) 0–11, (**b**) 12–19, and (**c**) 20+ days of age. Perches were placed in late perch (LP) treatment groups beginning at 18 weeks of age with rungs at heights pictured in (**c**).

**Figure 2 animals-14-00265-f002:**
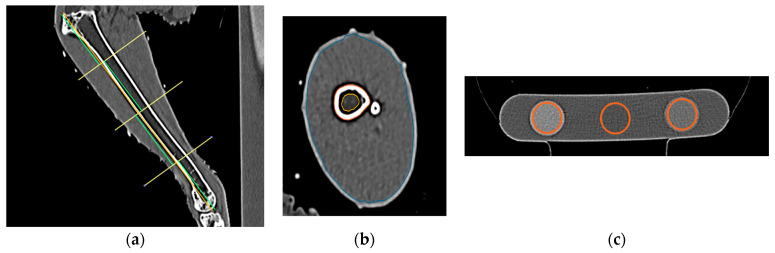
Steps of tibiotarsal image analysis. (**a**) Division of the tibiotarsus into 4 segments to set proximal, middle, and distal locations, (**b**) region of interest tracings for the tibiotarsus in the proximal location, and (**c**) region of interest placement in the 3 rods of hydroxyapatite phantom using the oval tool.

**Figure 3 animals-14-00265-f003:**
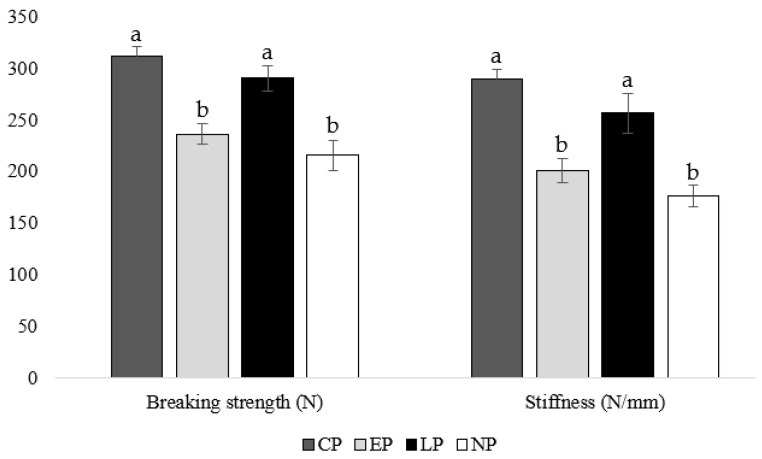
Mean tibia breaking strength (N) and stiffness (N/mm) of laying hens housed in continuous perch (CP), early perch (EP), late perch (LP), and no perch (NP) pens at 40 weeks of age (n = 84). ^a,b^ Means with differing superscripts indicate statistically significant differences between treatments within a parameter at *p* < 0.05.

**Figure 4 animals-14-00265-f004:**
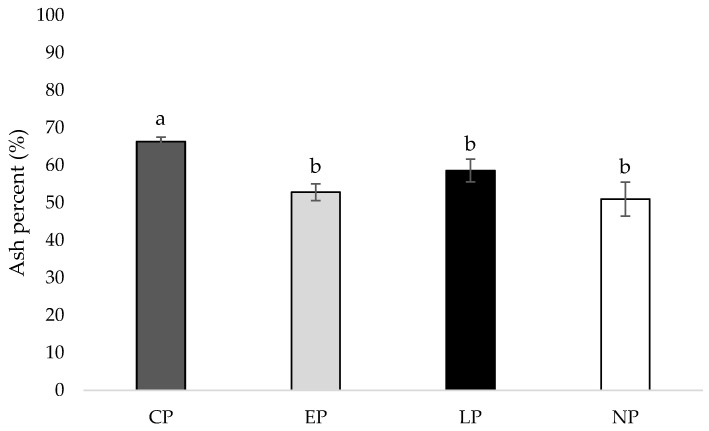
Mean tibia ash percent (%) of laying hens housed in continuous perch (CP), early perch (EP), late perch (LP), and no perch (NP) pens at 40 weeks of age (n = 84). ^a,b^ Means with differing superscripts indicate statistically significant differences between treatments at *p* < 0.05.

**Figure 5 animals-14-00265-f005:**
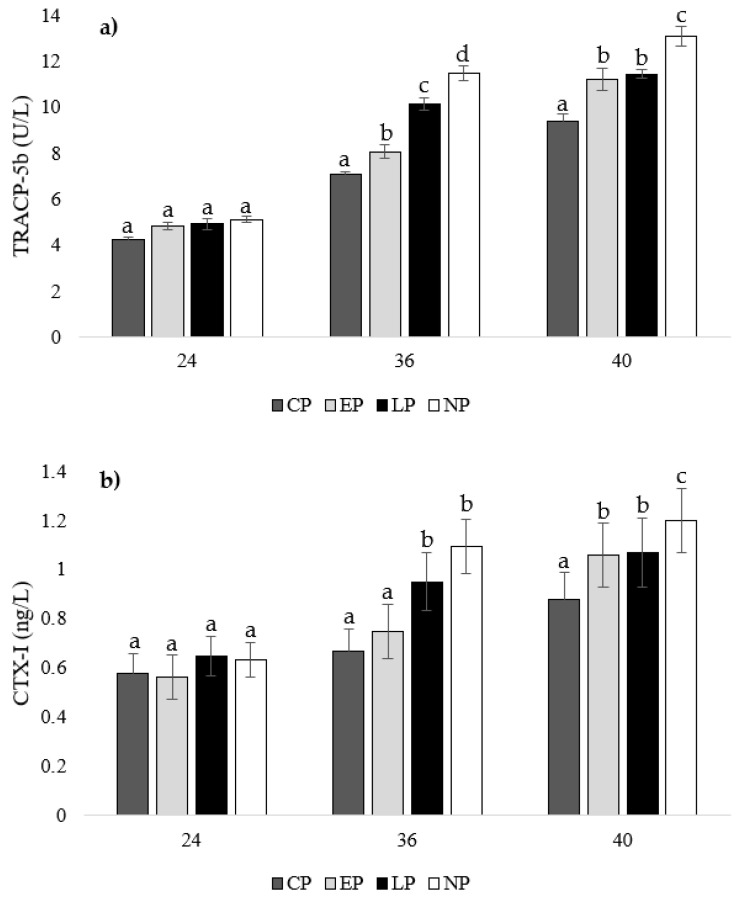
Mean serum concentrations of (**a**) tartrate-resistant acid phosphatase 5b (TRACP-5b; U/L) and (**b**) C-terminal telopeptide of type I collagen (CTX-I; ng/L) of laying hens housed in continuous perch (CP), early perch (EP), late perch (LP), and no perch (NP) pens at 24, 36, and 40 weeks of age (n = 84/week). ^a–d^ Means with differing superscripts indicate statistically significant differences between treatments within week at *p* < 0.05.

**Table 1 animals-14-00265-t001:** Overall, vertical, and horizontal activity levels, and average daily vertical displacement per bird (F) of laying hens housed in continuous perch (CP), early perch (EP), late perch (LP), and no perch (NP) pens at weeks 24, 36, and 40 of age (n = 84/week).

Week	Treatment	Overall Activity (g)	Vertical Activity (g)	Horizontal Activity (g)	Average Daily Vertical Displacement/Bird (F)
24	CP	1.42 ± 0.11 ^a^	0.59 ± 0.06 ^a^	0.83 ± 0.09 ^a^	24.52 ± 2.96 ^a^
	EP	1.33 ± 0.13 ^a^	0.18 ± 0.01 ^b^	1.15 ± 0.11 ^b^	3.69 ± 0.69 ^b^
	LP	1.39 ± 0.19 ^a^	0.51 ± 0.07 ^a^	0.88 ± 0.07 ^a^	23.58 ± 4.58 ^a^
	NP	1.29 ± 0.21 ^b^	0.11 ± 0.03 ^b^	1.18 ± 0.06 ^b^	1.13 ± 0.96 ^b^
	*p*-value	0.034	0.029	0.022	0.031
36	CP	1.44 ± 0.16 ^a^	0.56 ± 0.06 ^a^	0.88 ± 0.11 ^a^	33.25 ± 4.21 ^a^
	EP	1.45 ± 0.21 ^a^	0.12 ± 0.03 ^b^	1.33 ± 0.16 ^b^	3.56 ± 1.25 ^b^
	LP	1.41 ± 0.19 ^a^	0.55 ± 0.09 ^a^	0.86 ± 0.09 ^a^	29.87 ± 5.25 ^a^
	NP	1.31 ± 0.21 ^a^	0.11 ± 0.01 ^b^	1.20 ± 0.16 ^b^	2.03 ± 1.03 ^b^
	*p*-value	0.096	0.032	0.036	0.028
40	CP	1.35 ± 0.22 ^a^	0.53 ± 0.07 ^a^	0.82 ± 0.10 ^a^	28.85 ± 6.69 ^a^
	EP	1.39 ± 0.23 ^a^	0.13 ± 0.06 ^b^	1.26 ± 0.09 ^b^	4.03 ± 2.36 ^b^
	LP	1.36 ± 0.29 ^a^	0.59 ± 0.03 ^a^	0.77 ± 0.06 ^a^	26.85 ± 4.52 ^a^
	NP	1.32 ± 0.27 ^a^	0.09 ± 0.01 ^b^	1.23 ± 0.17 ^b^	1.63 ± 0.85 ^b^
	*p*-value	0.325	0.031	0.029	0.035

Treatments; CP: birds had continuous access to multi-tier perches from 0 to 40 weeks of age; NP: no access to perches from 0 to 40 weeks of age; EP: early access to perches during the rearing phase from 0 to 17 weeks of age; LP: Late access to perches during the laying phase from 17 to 40 weeks of age. ^a,b^ Means with differing superscripts indicate statistically significant differences within columns of the same week at *p* < 0.05.

**Table 2 animals-14-00265-t002:** Tibial total, medullary, and cortical bone mineral density (BMD; mg/cm^3^) and cross-sectional area (CSA; mm^2^) ±SEM for the proximal, middle, and distal regions of the right tibiotarsus of laying hens.

**Parameter/** **Treatment**	**Bone Cross-Sectional Area (mm^2^)**					
	**Total**			**Cortical**		
	**Proximal**	**Middle**	**Distal**	**Proximal**	**Middle**	**Distal**
CP	70.2 ± 1.1 ^a^	53.8 ± 1.3 ^a^	55.5 ± 0.7 ^a^	37.1 ± 2.4 ^a^	29.5 ± 2.1 ^a^	29.2 ± 1.8 ^a^
EP	70.1 ± 1.3 ^a^	53.9 ± 1.0 ^a^	55.3 ± 0.6 ^a^	30.7 ± 2.1 ^b^	23.9 ± 1.6 ^b^	24.0 ± 1.5 ^b^
LP	69.9 ± 1.1 ^a^	54.5 ± 0.8 ^a^	56.0 ± 0.7 ^a^	31.5 ± 2.1 ^b^	25.4 ± 1.6 ^b^	24.2 ± 1.5 ^b^
NP	69.9 ± 1.1 ^a^	54.9 ± 0.7 ^a^	56.0 ± 1.1 ^a^	26.3 ± 1.7 ^c^	21.6 ± 1.4 ^c^	20.9 ± 1.4 ^b^
*p*-value	0.235	0.185	0.635	0.021	0.019	0.024
**Parameter/** **Treatment**	**Bone Mineral Density (mg/cm^3^)**					
	**Total**			**Cortical**		
	**Proximal**	**Middle**	**Distal**	**Proximal**	**Middle**	**Distal**
CP	515.7 ± 13.7 ^a^	730.6 ± 11.0 ^a^	806.8 ± 11.0 ^a^	1028.8 ± 23.6 ^a^	1746.8 ± 16.9 ^a^	1370.7 ± 889.5 ^a^
EP	428.1 ± 15.4 ^a^	591.8 ± 22.5 ^b^	661.6 ± 14.6 ^b^	740.7 ± 20.9 ^b^	1257.7 ± 17.9 ^b^	1000.6 ± 22.0 ^b^
LP	438.4 ± 14.0 ^a^	628.3 ± 18.8 ^b^	669.7 ± 17.4 ^b^	761.3 ± 21.0 ^b^	1380.0 ± 22.6 ^b^	1165.1 ± 19.6 ^b^
NP	356.3 ± 11.4 ^b^	502.6 ± 13.1 ^c^	545.8 ± 20.7 ^c^	726.9 ± 23.3 ^c^	976.5 ± 19.1 ^c^	889.5 ± 23.1 ^c^
*p*-value	0.034	0.003	0.021	0.013	0.025	0.022

Treatments; CP: birds had continuous access to multi-tier perches from 0 to 40 weeks of age; NP: no access to perches from 0 to 40 weeks of age; EP: early access to perches during the rearing phase from 0 to 17 weeks of age; LP: Late access to perches during the laying phase from 17 to 40 weeks of age. ^a–c^ Means with differing superscripts indicate statistically significant differences within columns of the same week at *p* < 0.05.

**Table 3 animals-14-00265-t003:** Mean weight (g) ±SEM of biceps brachii, triceps brachii, pectoralis major, pectoralis minor, and leg muscle group of laying hens.

Treatment	Biceps Brachii (g)	Triceps Brachii (g)	Pectoralis Major (g)	Pectoralis Minor (g)	Leg Muscle Group (g)
CP	4.25 ± 0.26 ^a^	3.88 ± 0.31 ^a^	124.58 ± 4.85 ^a^	62.58 ± 5.55 ^a^	141.85 ± 7.98 ^a^
EP	3.65 ± 0.29 ^b^	3.59 ± 0.22 ^a^	112.55 ± 5.25 ^b^	56.85 ± 1.14 ^a^	124.55 ± 6.52 ^b^
LP	4.18 ± 0.38 ^a^	3.67 ± 0.29 ^a^	119.93 ± 4.99 ^a^	59.22 ± 3.55 ^a^	138.57 ± 5.88 ^a^
NP	3.52 ± 0.21 ^b^	3.53 ± 0.25 ^a^	107.58 ± 3.78 ^b^	55.85 ± 5.03 ^a^	120.79 ± 6.85 ^b^
*p*-value	0.026	0.259	0.031	0.523	0.028

Treatments: CP: birds had continuous access to multi-tier perches from 0 to 40 weeks of age; NP: no access to perches from 0 to 40 weeks of age; EP: early access to perches during the rearing phase from 0 to 17 weeks of age; LP: Late access to perches during the laying phase from 17 to 40 weeks of age. ^a,b^ Means with differing superscripts indicate statistically significant differences within columns of the same week at *p* < 0.05.

## Data Availability

For access to data from the study, please contact the corresponding author.

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
