# Peer review of "Impact of Perch Provision Timing on Activity and Musculoskeletal Health of Laying Hens"

_animals, 2024, doi:10.3390/ani14020265_

Round 1

Reviewer 1 Report

Comments and Suggestions for Authors

General comments:

 The manuscript titled “Impact of perch provision timing on activity and musculoskeletal health of laying hens” aimed to investigate the effect of perch provision to laying hens at different timing on bird activities, musculoskeletal health, and tibia breaking strength. The study design is appropriate, and methods were adequately described. The study demonstrate that authors dedicated extensive efforts to this study, collecting a substantial volume of data. The manuscript is well-written and easy to follow. However, there are areas throughout the manuscript that need clarification for readers. For example:

 ·         Bird activity was monitored over 3 consecutive days during weeks 24, 36, and 40 of 160 age. However, it’s not evident why the authors chose these specific time points to record bird activity. Please provide clarification.

·         Please include additional detailed information about the ELISA technique used for EAM. Specifically, provide the detection limit for each parameter measured, elaborate on how the standard curves were established, and explain the procedure for running samples in the 96 wells.

·         Considering that data were collected using 3 birds per pen, it is unclear how the data analysis was conducted. Did the authors average the data per pen?

·         Did the authors check the data for normality, considering that the logger was attached to the vest? How did the authors ensure that the obtained data were related to bird activity and not influenced by noise? Additionally, did the authors encounter technical issues, such as missing data from some data loggers?

·         The various types of bird activities should be clearly defined both in the Materials and Methods section and in the table. Additionally, it is unclear what the unit 'g' signifies. Furthermore, could you please clarify whether these activities are reported per day, per hour, or for the entire three days of the recording periods?

·         What is meant by 'vertical displacement'? This term has not been defined previously.

Specific comments: Please see attached pdf with side comments.

Author Response

Thank you for taking the time to review our manuscript. We appreciate the feedback. Responses to the comments are detailed below.

Bird activity was monitored over 3 consecutive days during weeks 24, 36, and 40 of 160 age. However, it’s not evident why the authors chose these specific time points to record bird activity. Please provide clarification.

AU: "The temporal selection of monitoring intervals at weeks 24, 36, and 40 was driven by a deliberate strategy aimed at comprehensively assessing avian behavioral patterns across key developmental milestones. To ensure a robust and nuanced understanding, this approach was designed to mitigate the influence of short-term fluctuations. Notably, environmental enrichments, specifically the introduction of perches, were provided to EP and CP birds from day 0, in contrast to LP hens that received perches at 17 weeks of age. This discrepancy allowed all birds sufficient time to adapt to perch use collectively. The decision to measure activity at 24 weeks of age (7 weeks after perch provision for LP hens) served the dual purpose of capturing a baseline assessment of perch-related behaviors and enabling the evaluation of potential late impacts on bird activity at weeks 36 and 40. Furthermore, these chosen time points align strategically with the production/laying cycle. Specifically, the 17-week mark coincides with hens being placed in the laying facility, while the onset of laying at 24 weeks corresponds to the period when egg production is peaking. Weeks 36 to 40 represent the peak lay period, a critical phase where hen production capacity is stretched to its maximum. In summary, the temporal framework for monitoring bird activity was intricately linked to both developmental and production-related milestones, ensuring a comprehensive and meaningful exploration of avian behaviors throughout the specified age range."

Please include additional detailed information about the ELISA technique used for EAM. Specifically, provide the detection limit for each parameter measured, elaborate on how the standard curves were established, and explain the procedure for running samples in the 96 wells.

AU: we appreciate your thoughtful reminder. While we did specify the names and manufacturers of the ELISA kits employed, it came to our attention that we inadvertently omitted the mention that the analyses were carried out following the manufacturer's protocol. We incorporated this statement intentionally to underscore adherence to the standard protocol endorsed by the manufacturer for conducting these tests. Our decision to abstain from detailing the procedure was a deliberate choice, aimed at preventing any potential issues related to plagiarism, given that the protocol is readily available online and is subject to the manufacturer's copyrights, alternatively we included methods of sampling and sample preparation.

Considering that data were collected using 3 birds per pen, it is unclear how the data analysis was conducted. Did the authors average the data per pen?

AU: GLMMs were performed utilizing the lme4 package, wherein individual birds nested within pens were incorporated as random effects within the statistical model (L318-330). This inclusion was vital to address the inherent variability associated with different bird-pen combinations. Consequently, the dataset encompassed individual bird performance metrics such as blood samples and activity, which were systematically collected and subsequently averaged per bird within each respective pen. This approach ensured a comprehensive analysis that accounted for both individual and collective variations in the observed parameters.

Did the authors check the data for normality, considering that the logger was attached to the vest? How did the authors ensure that the obtained data were related to bird activity and not influenced by noise? Additionally, did the authors encounter technical issues, such as missing data from some data loggers?

AU: thank you so much. Regarding your inquiry, we applied post-processing techniques to the logger data by implementing a 3-points average moving low-pass filter coupled with a step equation. This method was employed to effectively eliminate minor fluctuations or data noise attributed to factors unrelated to actual bird activity. Fortunately, this meticulous approach did not result in any data loss. It's worth noting that this data processing method has been utilized by the authors in previous studies. We not only adopted it for its efficacy but also took steps to validate and calibrate the method. For a comprehensive understanding of the validation and calibration procedures, please refer to the detailed information provided in the work by Ali and Siegford (2018) titled "An approach for tracking directional activity of individual laying hens within a multi-tier cage-free housing system (aviary) using accelerometers," published in Meas. Behav, 11, 176-180. The cited publication offers a thorough exposition of the methodology's robustness and reliability in tracking the directional activity of individual laying hens within a similar housing system. For the question about data normality; yes we have tested all the data for test residual distribution and assumptions for GLMM (L 322-330)

The various types of bird activities should be clearly defined both in the Materials and Methods section and in the table. Additionally, it is unclear what the unit 'g' signifies. Furthermore, could you please clarify whether these activities are reported per day, per hour, or for the entire three days of the recording periods?

AU: Thank you so much for the questions, g is a standardized measuring unit for acceleration, representing gravity, and can be expressed in units of m/s². description on the various types of activities were presented in the methods “Data on hens' vertical (az: dorsoventral movement across vertical levels), horizontal (ax: craniocaudal movement within the same vertical level), and lateral movement (ay: mediolateral movement within the same vertical level)”. Data were analyzed per bird within pen, and day of activity, and all were included as random effects to account for bird-pen and day variability; please see L321-322 for more details about the stats model “In each GLMM, the interaction term between main effects was also tested as fixed effects, and bird ID, pen, and day for activity”

What is meant by 'vertical displacement'? This term has not been defined previously.

AU: Thank you so much for capturing this. The statement clarifies the addition of information regarding vertical displacement in the methods section at L312.

Reviewer 2 Report

Comments and Suggestions for Authors

Thank you very much for the opportunity to review this paper on the impact of perch provision on musculoskeletal health in laying hens. Overall, the paper reads very well and it is clearly structured and easy to follow. With regards to the current keel bone damage issue in laying hens, I think this paper makes a good and relevant scientific contribution.

P1 L13: “from weeks 24 to 40” should be “from week 24 to 40”

P1 L20: I think your study allows for conclusions to be drawn about musculoskeletal health, but that you could only make assumptions about welfare since you don’t really know how these findings relate to keel bone damages

P1 L21: I think that you could be more specific here – you have shown that continuous access is better in terms of most of the parameters you’ve looked at (in comparison with no access in particular but also in comparison with early as well as late access)

P1 L28: “their” = “the hens’ musculoskeletal”

P1 L30-31: 7 pens per treatment = 28 pens. This does not correspond with the 30 pens as stated further down (P3 L120-121). Also, 28 pens with 29 birds per pen = 812 (not 810 as stated here). Further down (P3 L120-121) you have reported a total nr of 364 hens and 26 hens per pen?

P1 L37: Here you have reported a total of 84 hens, which does not correspond to what has been stated further down (P4 L181)?

P2 L51: hen shad = hens had

P2 L60: I presume this paper is written from an American perspective – could you provide one or two sentences regarding legislative requirements? In Sweden (where I am from), it is required by law to provide pullets with perches during rearing, although it is not according to EU regulations. I think the legislative background would add nicely to the introduction

P2 L64: I presume you mean medullary bone rather than structural bone as a calcium source, but this sentence reads like medullary bone is formed instead of structural bone altogether

P2 L68: not sure “experience” is the best word to use here

P2 L74-86: several of the sentences here does not read very well (quite long sentences with a lot of commas) and I especially suggest that you consider the use of “therefore” on e.g. L76, 78 and 85

P2 L84: crucial for what? Laying hen welfare?

P2 L89-92: I do not agree with the statement that “keel bone damage is typically assumed to be caused by collisions with furniture within the environment” (this theory has been refuted by more recent research although it collisions can of course not be completely left out of the aetiological equation). I think this background information needs to include some more recent research (there is plenty!) – reference nr 27 and 28 are from 1992 (and this is a conference abstract…) and 2004! Moreover, although “the exact factors influencing keel bone damage in laying hens are unknown” we do know that e.g. the early onset of lay and production of large eggs contribute, and that there seems to be a genetic component.

P2 L96: this sentence does not read very well

P3 L120-121: see my earlier comments about nr of birds and pens

P3 L123 + L127: no need to state that birds had ad libitum access to water and feed twice

P3 L135: this is a quite long dark period (compared to what is normally practiced at least) – why/how did you decide upon this light schedule?

P4 L157: when did you change the perch rungs? According to the explanation on L 149 – 151 it seems like it should be a) 0-11, b) 12-19

P4 L157: should “later perch” be “late perch”? 

P4 L160: I suggest you include the information about the accelerometer to this first sentence. I was a bit confused when reading the second sentence about catching the birds, since I first assumed that the activity levels were recorded through behavioural observations – so an explanation as to why you would catch them would be good earlier on here I think

P4 L175: how did you ensure this i.e. what behaviours/indicators did you look at?

P4 L181: why were the birds killed using CO2? It is well known that CO2 is aversive and there are more humane ways of killing poultry e.g. cervical dislocation after stunning. Although 90 birds might be considered quite many, it is still not too many for manual handling of each bird to be feasible, and it wouldn’t have impaired your post mortem analysis?

P4 L205: could you include one sentence explaining what this tool is/does (not something everyone is familiar with)

P6 L215: “below the bottom of the keel bone” – where is this? Caudal to the keel bone tip?

P6 L240: could you include a short explaining sentence of the purpose of the saline-soaked paper?

P7 L255: for consistency, I suggest you include this information also in the previous section about the right tibia

P7 L257: not completely clear whether this refers to the tibiae or fibula

P7 L268: was this the same/different birds from each pen at each age? Or could it have been?

P7 L270: I would have appreciated one or two sentences to explain what these measures are i.e. what does a high/low value tell us

P8 L295-297: these video recordings have not been mentioned earlier (e.g. under 2.4.) – was this done throughout the experiment or just for a short period of time/some birds to obtain enough material to be able to do this calibration you describe here?

P8 L325: is this “average number of daily displacements per bird”?

P9 Table 1: what units are (g) and (F) here?

P9 L339: this is a very long sentence – suggest you break it in two after “Table 2)”

P9 L345: this is a very long sentence – suggest you break it in two after “EP and LP.”

P10 Table 2: it seems superfluous to have the same information both in the heading and below the table.

P10 L356: I would suggest to use “heavier” instead of “larger” since you only weighed them and thus technically only have the weight, not the size

P10 Table 3: it seems superfluous to have the same information both in the heading and below the table.

P11 Table 6: is missing

P12 L406-407: why do you think that is/could you elaborate/speculate on that a bit?

P12 L406-409: the NP and EP hens showed more horizontal movements than CP and LP, right? Could this perhaps reflect a frustration/displacement behaviour in the hens since they did not have access to perches?

P12 L408-409: I miss a, perhaps quite obvious, sentence about the lack of perches in NP and EP which inevitably means that they did not have the same opportunity/reason/motivation to move vertically i.e. they could not move vertically to the same extent

P12 L413-414: (How) was this behaviour registered? Or is it just “personal observations” (which then should be clarified)

P13 L424-425: This sentence appears a little “odd” here – between your own results and previous studies to confirm this and the conclusion drawn from these results. I would suggest that 1. You move this to after “These results suggest that… …compared to no perch access at all.” And 2. You add a sentence about possible plausible explanations for this difference?

P13 L426: Does “these” refer to your own results or to the aforementioned other studies?

P13 L431: What is this “conversely” to?

P13 L436: see my earlier comment regarding the use of “larger” vs “heavier”

P13 L442-445: I don’t really see how LP > EP muscle weight is contrary to the previous work showing that early access to perches = increased muscle deposition in adults? Your results show that late perch access can increase MORE than if only early access…? However, to me it seems that your results showing no difference between EP and NP would be in contrast to previous work (15,39) since you would otherwise have expected muscle weight EP > NP?

P13 L452-464: this first part of section 4.3 does not read very well – it is a bit hard to follow with all the different studies showing this and that. Also, this is mostly a short review of previous studies that does not really relate to your own findings – could you explain/speculate/elaborate a bit on what these different results across studies might be due to?

P14 L484-485: reading this, I am left curious to know why tibia ash content was not higher in EP (does it decline after X weeks?) or LP (does it take X weeks before an increase can be seen?) Could you elaborate on this?

P14 L486: see my comment on P7 L270 (for someone who is not an expert in this it would be really helpful to have one or two sentences to explain what these measures are i.e. what does a high/low value tell us)

P14 L509: “exerts a favorable influence on the musculoskeletal health and activity levels of laying 509 hens” – isn’t is that perches led to a higher activity level, which in turn led to improved musculoskeletal health? It seem to me that this is the reasoning that has been put forth through the paper

P15 L510-511: “thereby contributing to an improvement in overall hen welfare” – could you perhaps link this to your introduction about keel bone damages (to make it explicit what you mean with better welfare)? Also, I think one of your main findings is that continuous perch access seem to be necessary to obtain the best possible improvements in terms of musculoskeletal health, which you have indeed mentioned here, but which I think you should emphasise even more. 

Author Response

Thank you for your kind words and thorough feedback on our manuscript. It is much appreciated.

P1 L13: “from weeks 24 to 40” should be “from week 24 to 40”

AU: thank you so much this has been amended.

P1 L20: I think your study allows for conclusions to be drawn about musculoskeletal health, but that you could only make assumptions about welfare since you don’t really know how these findings relate to keel bone damages.

AU: we totally agree, and the text was revised as per your suggestions

P1 L21: I think that you could be more specific here – you have shown that continuous access is better in terms of most of the parameters you’ve looked at (in comparison with no access in particular but also in comparison with early as well as late access).

AU: thank you so much, we added some specificity in the concluding statements in L21-22.

P1 L28: “their” = “the hens’ musculoskeletal”.

AU: This has been edited.

P1 L30-31: 7 pens per treatment = 28 pens. This does not correspond with the 30 pens as stated further down (P3 L120-121). Also, 28 pens with 29 birds per pen = 812 (not 810 as stated here). Further down (P3 L120-121) you have reported a total nr of 364 hens and 26 hens per pen?

P1 L37: Here you have reported a total of 84 hens, which does not correspond to what has been stated further down (P4 L181)?

AU: Thank you sincerely for addressing this matter, and we deeply value the meticulous revisions you have made. The numerical discrepancies have been rectified, and the accurate count of total birds utilized is now established at 812, distributed across seven pens per treatment and amounting to a total of 28 pens, each housing 29 birds. It is noteworthy that, initially, 840 day-old chicks were employed at the commencement of the trial, distributed among 30 pens (15 with or without perches) until reaching 17 weeks of age. Upon reaching the 17-week mark, the bird population was reassessed, and after factoring in mortalities, a total of 812 birds remained. Subsequently, these birds were appropriately redistributed based on the previously outlined four treatment conditions, as detailed in P1 L30-31, P3 L120-121, and P3 L120-121. Your attention to detail is greatly appreciated, and we express our gratitude for your thoroughness in capturing and rectifying these adjustments.

P2 L51: hen shad = hens had.

AU: Thank you for catching this error, it has been edited.

P2 L60: I presume this paper is written from an American perspective – could you provide one or two sentences regarding legislative requirements? In Sweden (where I am from), it is required by law to provide pullets with perches during rearing, although it is not according to EU regulations. I think the legislative background would add nicely to the introduction.

AU: thank you for your kind suggestion, a statement describing the situation was added to the text

P2 L64: I presume you mean medullary bone rather than structural bone as a calcium source, but this sentence reads like medullary bone is formed instead of structural bone altogether.

AU: This statement has been clarified.

P2 L68: not sure “experience” is the best word to use here.

AU: Agreed, this has been deleted.

P2 L74-86: several of the sentences here does not read very well (quite long sentences with a lot of commas) and I especially suggest that you consider the use of “therefore” on e.g. L76, 78 and 85.

AU: This paragraph has hopefully been clarified by reducing sentence length.

P2 L84: crucial for what? Laying hen welfare?

AU: Word choice has been edited.

P2 L89-92: I do not agree with the statement that “keel bone damage is typically assumed to be caused by collisions with furniture within the environment” (this theory has been refuted by more recent research although it collisions can of course not be completely left out of the aetiological equation). I think this background information needs to include some more recent research (there is plenty!) – reference nr 27 and 28 are from 1992 (and this is a conference abstract…) and 2004! Moreover, although “the exact factors influencing keel bone damage in laying hens are unknown” we do know that e.g. the early onset of lay and production of large eggs contribute, and that there seems to be a genetic component.

AU: totally agree, thank you for pointing this out. Background information has been added and updated references included.

P2 L96: this sentence does not read very well.

AU: This has been amended.

P3 L120-121: see my earlier comments about nr of birds and pens

AU; thank you, this is corrected now

P3 L123 + L127: no need to state that birds had ad libitum access to water and feed twice. Thanks for catching this, it is now only included once.

P3 L135: this is a quite long dark period (compared to what is normally practiced at least) – why/how did you decide upon this light schedule?

AU: We followed the breed guidelines for alternative housing systems. chrome-extension://efaidnbmnnnibpcajpcglclefindmkaj/https://www.hyline.com/filesimages/Hy-Line-Products/Hy-Line-Product-PDFs/Brown/BRN%20COM%20ENG.pdf

P4 L157: when did you change the perch rungs? According to the explanation on L 149 – 151 it seems like it should be a) 0-11, b) 12-19.

AU: Thank you for catching this inconsistency, the dates have been edited.

P4 L157: should “later perch” be “late perch”? 

AU: Yes, thank you!

P4 L160: I suggest you include the information about the accelerometer to this first sentence. I was a bit confused when reading the second sentence about catching the birds, since I first assumed that the activity levels were recorded through behavioural observations – so an explanation as to why you would catch them would be good earlier on here I think.

AU: explanation was added to the first sentence as per your suggestion

P4 L175: how did you ensure this i.e. what behaviours/indicators did you look at?

AU: accelerometers have clear orientation guidelines for how to adjust the directions on the animal body to ensure directional activity matches with the correct animal movements. 

P4 L181: why were the birds killed using CO2? It is well known that CO2 is aversive and there are more humane ways of killing poultry e.g. cervical dislocation after stunning. Although 90 birds might be considered quite many, it is still not too many for manual handling of each bird to be feasible, and it wouldn’t have impaired your post mortem analysis?

AU: We completely agree with the preference for cervical dislocation (CD) and the reduction in sample size. Our decision to forego CD was primarily motivated by the upcoming CT scans post-euthanasia, as we sought to avoid potential body damage resulting from handling or convulsions during the dislocation process. Given that CO2 euthanasia is an AVMA-approved method and adheres to farm SOPs, we opted for this approach to minimize convulsions and reduce the risk of associated body damage. This choice aligns with our goal of preserving the integrity of post-mortem imaging. In terms of sample size considerations, employing a total of 84 birds with 3 birds sampled from each pen (out of the 29 birds per pen) amounts to approximately 10%. This proportion is considered sufficient for detecting potential significant differences across treatments. Our decision reflects a careful balance between obtaining robust data and minimizing the impact on the avian subjects.

P4 L205: could you include one sentence explaining what this tool is/does (not something everyone is familiar with)

AU: We are sorry but we think the line order was slightly shifting, If the reference is to the calibration phantom utilized in the CT scanning, it is pertinent to note that we have appropriately cited our previously published technical study. This comprehensive publication provides detailed elucidation of all the procedures and equipment employed throughout this process.

P6 L215: “below the bottom of the keel bone” – where is this? Caudal to the keel bone tip?

AU: Yes, the cut was made on the caudal tip of the keel bone. This has been clarified.

P6 L240: could you include a short explaining sentence of the purpose of the saline-soaked paper?

AU: Of course, an explanation has been added.

P7 L255: for consistency, I suggest you include this information also in the previous section about the right tibia.

AU: information is now included in L238-241.  

P7 L257: not completely clear whether this refers to the tibiae or fibula.

AU: This has been clarified.

P7 L268: was this the same/different birds from each pen at each age? Or could it have been?

AU: We employed distinct birds at each timepoint to encompass the maximum feasible number of subjects throughout the study, thereby enhancing the representativeness of our sample.

P7 L270: I would have appreciated one or two sentences to explain what these measures are i.e. what does a high/low value tell us

AU: We are sorry but we think the line order was slightly shifting, if you are referring to bone demineralization measures: To assess the magnitude of bone resorption, various biochemical markers are commonly utilized in clinical practice. C-terminal telopeptide of type I collagen (CTX-I) and tartrate-resistant acid phosphatase 5b (TRACP-5b), as previously mentioned, are two such markers that provide valuable insights into bone metabolism. To comprehensively evaluate the extent of bone resorption, a combination of these markers along with other diagnostic tools can be employed. A statement about the use of these lab tests was added

P8 L295-297: these video recordings have not been mentioned earlier (e.g. under 2.4.) – was this done throughout the experiment or just for a short period of time/some birds to obtain enough material to be able to do this calibration you describe here?

AU: Video recording was systematically carried out during the entire study duration to gather data for behavior observation, which is intended for a separate publication. However, as previously mentioned, a subset of these recorded videos was utilized for calibration purposes, as elaborated upon in your explanation.

P8 L325: is this “average number of daily displacements per bird”?

AU: yes and explanation was added

P9 Table 1: what units are (g) and (F) here?

AU: explanation was added in the method section: g is a standardized measuring unit for acceleration, representing gravity, and can be expressed in units of m/s². description on the various types of activities were presented in the methods “Data on hens' vertical (az: dorsoventral movement across vertical levels), horizontal (ax: craniocaudal movement within the same vertical level), and lateral movement (ay: mediolateral movement within the same vertical level)”.

P9 L339: this is a very long sentence – suggest you break it in two after “Table 2)”

AU: This has been edited.

P9 L345: this is a very long sentence – suggest you break it in two after “EP and LP.”

AU: This has been edited.

P10 Table 2: it seems superfluous to have the same information both in the heading and below the table.

AU: we totally agree, data were removed from the heading

P10 L356: I would suggest to use “heavier” instead of “larger” since you only weighed them and thus technically only have the weight, not the size.

AU: Agreed, this has been changed.

P10 Table 3: it seems superfluous to have the same information both in the heading and below the table.

AU: we totally agree, data were removed from the heading

P11 Table 6: is missing

AU: thank you, we believe there was an error in the file upload process, figure 6 is missing but was added to the revised version of the manuscript.

P12 L406-407: why do you think that is/could you elaborate/speculate on that a bit?

P12 L406-409: the NP and EP hens showed more horizontal movements than CP and LP, right? Could this perhaps reflect a frustration/displacement behaviour in the hens since they did not have access to perches?

AU: We concur that the observed behavior might be attributed to the absence of perches, hindering the performance of perching behavior. In the absence of perches, birds tend to modify their daily activity repertoire. Nevertheless, it is noteworthy that the overall activity levels across treatments were not significantly affected. It is acknowledged that, given access to perches, birds typically exhibit a preference for engaging in perching behavior as part of their daily activities.

P12 L408-409: I miss a, perhaps quite obvious, sentence about the lack of perches in NP and EP which inevitably means that they did not have the same opportunity/reason/motivation to move vertically i.e. they could not move vertically to the same extent.

AU; Yes, this is a sentence that should be included regardless of if it is obvious. Thank you!

P12 L413-414: (How) was this behaviour registered? Or is it just “personal observations” (which then should be clarified).

AU: Yes, this was observed anecdotally and has been included within the sentence.

P13 L424-425: This sentence appears a little “odd” here – between your own results and previous studies to confirm this and the conclusion drawn from these results. I would suggest that 1. You move this to after “These results suggest that… …compared to no perch access at all.” And 2. You add a sentence about possible plausible explanations for this difference? AU: Agreed, thank you for these suggestions. They have been incorporated.

P13 L426: Does “these” refer to your own results or to the aforementioned other studies?

AU: This has been clarified.

P13 L431: What is this “conversely” to?

AU: This has been amended.

P13 L436: see my earlier comment regarding the use of “larger” vs “heavier”

AU: Agreed, thank you!

P13 L442-445: I don’t really see how LP > EP muscle weight is contrary to the previous work showing that early access to perches = increased muscle deposition in adults? Your results show that late perch access can increase MORE than if only early access…? However, to me it seems that your results showing no difference between EP and NP would be in contrast to previous work (15,39) since you would otherwise have expected muscle weight EP > NP?

AU: we totally agree, we were expecting heavier muscles in EP than NP that would be statistically significant, however, after data analysis it seems that EP showed heaver muscles than NP but unfortunately the differences were not statistically significant, we are working on analysing the data during the rearing phase to better understand these findings. On the other hand, our results indicate that even after sexual maturity perch provision still can improve musculoskeletal health. 

P13 L452-464: this first part of section 4.3 does not read very well – it is a bit hard to follow with all the different studies showing this and that. Also, this is mostly a short review of previous studies that does not really relate to your own findings – could you explain/speculate/elaborate a bit on what these different results across studies might be due to?

P14 L484-485: reading this, I am left curious to know why tibia ash content was not higher in EP (does it decline after X weeks?) or LP (does it take X weeks before an increase can be seen?) Could you elaborate on this?

AU: previous work focused mainly on the influence of perch provision/increased activity on musculoskeletal health during rearing phase, with some studies tried to expand the testing to the lay phase. In our study our EP group were not given access to perches starting at 17 weeks of age, after been able to perch during the rearing phase <17 woa. Unlike the LP hens that only had access to perches in the laying phase to be a direct contrast to EP group. In our study, however since our trial ended at 40 woa, it seems like the effect of perch provision during the x (40-17 = 23) weeks had better effect on improving musculoskeletal heath that the early 17 weeks of age (EP hens). Although, these findings are not in consistent with our hypothesis, however, it highlights the great opportunity lies on perch provision during lay phase on improving musculoskeletal health of hens, even if perches were not provided during the rearing phase. We totally agree that we mentioned a lot of previous studies while trying to give the readers the opportunity to align our findings with previous studies and elaborate that our findings may be different than what was published/expected.   

P14 L486: see my comment on P7 L270 (for someone who is not an expert in this it would be really helpful to have one or two sentences to explain what these measures are i.e. what does a high/low value tell us).

AU: Higher values indicate higher rates of bone demineralization/resorption, whereas low values indicate lower levels of bone demineralization (i.e., “better”).

P14 L509: “exerts a favorable influence on the musculoskeletal health and activity levels of laying 509 hens” – isn’t is that perches led to a higher activity level, which in turn led to improved musculoskeletal health? It seem to me that this is the reasoning that has been put forth through the paper.

AU: Yes, this is a valid point, this has been altered within the conclusion.

P15 L510-511: “thereby contributing to an improvement in overall hen welfare” – could you perhaps link this to your introduction about keel bone damages (to make it explicit what you mean with better welfare)? Also, I think one of your main findings is that continuous perch access seem to be necessary to obtain the best possible improvements in terms of musculoskeletal health, which you have indeed mentioned here, but which I think you should emphasise even more.

AU: Yes,  the connection was highlighted in the introduction

Reviewer 3 Report

Comments and Suggestions for Authors

General comments:

Figure 2 is the same figure as figure 1 in a previous article (https://doi.org/10.2460/ajvr.23.05.0109).  The same applies to figure 4, which is the same as figure 2 in that publication. This means that the figures are not original and could be a copyright issue.

Including bone weight and relative bone weight (as a percentage of total body weight) could provide valuable insights into the skeletal impact of perch access. Similarly, presenting relative muscle weight could augment the understanding of muscle development in different groups.

As CT scans are available, providing cross-section images of bone mid-diaphysis from CT scans could visually demonstrate potential changes in medullary bone formation in the tibias of hens from different groups.

The study would benefit from acknowledging certain limitations, such as the exclusion of wing bone analysis. The perch activity likely influences the bones of the chest, shoulder girdle, and wings, as suggested by the changes in pectoral muscle mass. Additionally, the Introduction highlights keel bone features, yet the study lacks an analysis of this skeletal part. Furthermore, only bone resorption markers were analyzed (markers of osteoclast activity), while no markers of osteoblast activity are presented.

Specific comments:

L87-92 the sentence “the exact factors influencing keel bone damage in laying hens are unknown” seems to contradict the earlier statement about injuries being caused by collisions. Clarification or additional context could resolve this apparent inconsistency.

Introduction: Please consider briefly explaining the rationale behind the selection of the tibia over the femur for this study.

L131 verify whether 21 degrees is Fahrenheit or Celsius.

L137 It would be helpful to explain why these specific weeks (24, 36, and 40) were chosen for monitoring.

L191 Could you specify the average duration of a CT scan for a single bird?

L249 in a three-point bending test, bending moment is equal to FL/4, where F is bending stiffness and L is span (named furculum in this study) width. As the latter was the same for all samples (1 cm), the bending moment is just a bending stiffness F multiplied by 0.01 m. Therefore, inclusion of bending moment to the results does not bring any new data to the study. Interestingly, like referring to my comment a priori, the bending moment results are missing in the results section.

L226 Consider using the term “bone resorption markers” for clarity.

L278-297 ax,az,ay, As – subscripts

L289 verify the equation. A_j A'_i = μ when |A_i - μ| < t. For |A_i - μ| ≥ t, the result should be different, probably A_j A'_i = A_i.

L310 stiffness (N/mm)

Table 1 Please clarify the unit of daily vertical displacement, as “F” is not a standard unit in this context. "F" is also the unit for farad.

Figure 5. Verify the accuracy of the figure, particularly whether the bone bending before fracture was approximately 1 mm for all samples.

Figure 6 is missing.

Discussion: In the Discussion section, it would be beneficial for the authors to compare their study findings with the practices of commercial egg production, particularly regarding the accessibility of perches in these facilities. Such a comparison would aid in translating the research findings into practical applications within the industry, potentially leading to improvements in the welfare of laying hens in a commercial setting.

Author Response

Thank you for taking the time to review our manuscript. We appreciate the feedback. Responses to the comments are detailed below.

Figure 2 is the same figure as figure 1 in a previous article (https://doi.org/10.2460/ajvr.23.05.0109).  The same applies to figure 4, which is the same as figure 2 in that publication. This means that the figures are not original and could be a copyright issue.

AU: thank you so much for capturing this, this is a technical study published by our team focusing mainly on CT scanning procedures and analysis, since we cited our previous work we decided to delete these figures from the current study.

Including bone weight and relative bone weight (as a percentage of total body weight) could provide valuable insights into the skeletal impact of perch access. Similarly, presenting relative muscle weight could augment the understanding of muscle development in different groups.

AU: We fully concur with your perspective. Nevertheless, our decision to present ash% as an indicator for the relative ash weight of the Tibiotarsal bone is driven by the belief that it better reflects bone health comparisons across groups. Regarding muscle weight, while acknowledging that presenting relative muscle weight would offer a more accurate indicator for comparing muscle development across treatments, we have opted to present the absolute fully dissected muscle weight. This decision serves two primary purposes: first, to provide reference values for the average weight of these muscles in the utilized breed of laying hens, and second, to potentially enhance the comparability of these findings across studies by presenting the average and standard deviation of muscle weights.

As CT scans are available, providing cross-section images of bone mid-diaphysis from CT scans could visually demonstrate potential changes in medullary bone formation in the tibias of hens from different groups.

AU: thank you so much for wonderful suggestion, we focused on presenting total and cortical CSA in the current study, as we are working on investigating the influence of age/activity/production on medullary CSA and density using the same set of CT scans. 

The study would benefit from acknowledging certain limitations, such as the exclusion of wing bone analysis. The perch activity likely influences the bones of the chest, shoulder girdle, and wings, as suggested by the changes in pectoral muscle mass. Additionally, the Introduction highlights keel bone features, yet the study lacks an analysis of this skeletal part. Furthermore, only bone resorption markers were analyzed (markers of osteoclast activity), while no markers of osteoblast activity are presented.

AU: thank you so much, in the current study we used Tibiotarsal bone as an indicator for bone health across treatments, in another published study by our group we focused on the intercorrelation between muscle deposition and bone density. Regarding the wing bone assessment, we are planning to test the influence of wing assisted jumping and flying incidence on wing bone health suing the same set of data, but correlate that with the incidence of wing activity.

Specific comments:

L87-92 the sentence “the exact factors influencing keel bone damage in laying hens are unknown” seems to contradict the earlier statement about injuries being caused by collisions. Clarification or additional context could resolve this apparent inconsistency.

AU: thank you so much, clarification was added to the text as per your suggestions.

Introduction: Please consider briefly explaining the rationale behind the selection of the tibia over the femur for this study.

AU: Tibia has been selected by several studies focusing mainly on investigating bone health in laying hens, therefore, one reason it was selected for the current study is to make our findings comparable to other studies within the same field, Anatomically, tibia is more suitable for mechanical testing than femur due to the more flattened ventromedial surface compared to femur. It was easier to obtain a valid CT scan for the tibia rather than femur without any superimposition or artifacts, and it was easier for us to divide it into 3 anatomically equal portions for CT analysis. 

L131 verify whether 21 degrees is Fahrenheit or Celsius.

Au: thank you so much for capturing this, it was corrected

L137 It would be helpful to explain why these specific weeks (24, 36, and 40) were chosen for monitoring.

AU: The temporal selection of monitoring intervals at weeks 24, 36, and 40 was driven by a deliberate strategy aimed at comprehensively assessing avian behavioral patterns across key developmental milestones. To ensure a robust and nuanced understanding, this approach was designed to mitigate the influence of short-term fluctuations. Notably, environmental enrichments, specifically the introduction of perches, were provided to EP and CP birds from day 0, in contrast to LP hens that received perches at 17 weeks of age. This discrepancy allowed all birds sufficient time to adapt to perch use collectively. The decision to measure activity at 24 weeks of age (7 weeks after perch provision for LP hens) served the dual purpose of capturing a baseline assessment of perch-related behaviors and enabling the evaluation of potential late impacts on bird activity at weeks 36 and 40. Furthermore, these chosen time points align strategically with the production/laying cycle. Specifically, the 17-week mark coincides with hens being placed in the laying facility, while the onset of laying at 24 weeks corresponds to the period when egg production is peaking. Weeks 36 to 40 represent the peak lay period, a critical phase where hen production capacity is stretched to its maximum. In summary, the temporal framework for monitoring bird activity was intricately linked to both developmental and production-related milestones, ensuring a comprehensive and meaningful exploration of avian behaviors throughout the specified age range.

L191 Could you specify the average duration of a CT scan for a single bird?

Au: information was added to the text as per your suggestion

L249 in a three-point bending test, bending moment is equal to FL/4, where F is bending stiffness and L is span (named furculum in this study) width. As the latter was the same for all samples (1 cm), the bending moment is just a bending stiffness F multiplied by 0.01 m. Therefore, inclusion of bending moment to the results does not bring any new data to the study. Interestingly, like referring to my comment a priori, the bending moment results are missing in the results section.

AU: Thank you so much for your valuable inputs, our calculations for stiffness was as follows: stiffness in our 3 point bending testing is determined by measuring the slope of the linear portion of the force-deflection curve. The stiffness is calculated as the ratio of the applied force to the resulting displacement or deflection k = force/displacement (N/mm) expressed in units of force per unit of displacement, such as Newtons per millimeter (N/mm: expressed in units of force per unit of deformation, such as Newtons per millimeter). A stiffer bone will exhibit less deformation under a given load compared to a less stiff bone.

L226 Consider using the term “bone resorption markers” for clarity.

AU: thank you so much for the suggestion, this was changed as per your recommendation.

L278-297 ax,az,ay, As – subscripts

AU: thank you so much for the suggestion, this was changed as per your recommendation

L289 verify the equation. A_j A'_i = μ when |A_i - μ| < t. For |A_i - μ| ≥ t, the result should be different, probably A_j A'_i = A_i.

AU: Thank you for providing your valuable input. It appears that Microsoft Word may have misplaced the symbols. If you meant 'Ai' to represent the new middle point outcome of the 3-point low-pass filter after applying the cut-off points of the step function, then it signifies the new data point after smoothing. I hope this clarification resolves any confusion

L310 stiffness (N/mm)

AU: thank you so much for the suggestion, this was changed as per your recommendation

Table 1 Please clarify the unit of daily vertical displacement, as “F” is not a standard unit in this context. "F" is also the unit for farad.

AU: thank you so much for the suggestion, this was changed as per your recommendation, and clarification was added to the method section as well.  

Figure 5. Verify the accuracy of the figure, particularly whether the bone bending before fracture was approximately 1 mm for all samples.

AU: thank you, we have verified the figure as per your suggestion, the bending was variable as we explained earlier we calculated the stiffness in our 3 point bending testing is determined by measuring the slope of the linear portion of the force-deflection curve. The stiffness is calculated as the ratio of the applied force to the resulting displacement or deflection k = force/displacement (N/mm) expressed in units of force per unit of displacement, such as Newtons per millimeter (N/mm: expressed in units of force per unit of deformation, such as Newtons per millimeter). A stiffer bone will exhibit less deformation under a given load compared to a less stiff bone.

Figure 6 is missing.

AU: thank you so much, for some reason this figure was not shown In the submission, it is added now, and the figure order was reconsidered after deleting some figures as per your suggestion.

Discussion: In the Discussion section, it would be beneficial for the authors to compare their study findings with the practices of commercial egg production, particularly regarding the accessibility of perches in these facilities. Such a comparison would aid in translating the research findings into practical applications within the industry, potentially leading to improvements in the welfare of laying hens in a commercial setting.

AU: the discussion was improved and some information were added, we truly appreciate all your valuable inputs and help  improving our manuscript